# Conjecture extraction for proof autoformalization using large language models and automated theorem provers

## Abstract

Autoformalization and ATP have each advanced the mechanization of mathematics, yet the translation of informal proofs into fully formalized counterparts remains an open challenge: especially for interactive theorem provers beyond Isabelle. We introduce conjecture extraction, a novel proof autoformalization pipeline tailored to Lean 4 that decomposes an informal proof into individual lemmas (conjectures), formalizes and proves each in isolation, and then reassembles them to recover the original argument. Unlike prior sketch-based methods, our approach is compatible with end-to-end proof generation models and leverages repeated conjecture refinement to incrementally improve performance. We implement an open-source system that integrates off-the-shelf autoformalization LLMs, automated theorem provers, and an online reinforcement-learning loop to optimize both conjecture generation and proof search. On the MiniF2F benchmark, conjecture extraction achieves an absolute improvement of 11.2 percentage points in pass@1 over the Draft, Sketch, and Prove port (DSP) for Lean 4, demonstrating the efficacy of proof decomposition and recomposition. Our results suggest that conjecture extraction not only bridges a gap in proof autoformalization for Lean but also offers a general framework for scaling formalization efforts across diverse proof assistants. We release our code and models to foster further research in large-scale formalized mathematics.

## 1 Introduction

Formalization is costly, requiring significant labor by a small group of human experts. To aid mathematicians and enable large-scale formalization, autoformalization has been introduced Wu et al. (2022). Autoformalization translates informal statements into formal versions using Large Language Models (LLMs) Wu et al. (2022). Orthogonal to autoformalization, Automated Theorem Proving (ATP) attempts to prove a formal conjecture. Combinations of autoformalization and ATP focus on large-scale conjecture autoformalization to enable more potential training data for ATP models Wu et al. (2024a).

Few works focus on proof autoformalization, translating existing informal proofs into their formalized counterparts Jiang et al. (2023). This area is especially relevant for research mathematics, where current ATP solutions still fall short. Works on proof autoformalization target the proof assistant Isabelle Paulson (1994). Transferring this to Lean 4 Moura & Ullrich (2021) is difficult Lin et al. (2025). The only successful implementation of proof autoformalization is, to the best of our knowledge, a port of *Draft, Sketch, and Prove* (DSP) Jiang et al. (2023) by Aniva et al. (2025). In this work, we propose *conjecture extraction*. We decompose proofs into individual proof steps, prove them individually, and recompose the overarching proof. As our setup does not rely on proof sketching, it is compatible with whole-proof generation models. Further, we improve over DSP for Lean, and show how repeated conjecture extraction further boosts performance.

Our contributions are as follows:

- We present conjecture extraction, a proof autoformalization method designed for Lean 4, compatible with whole-proof generation models.

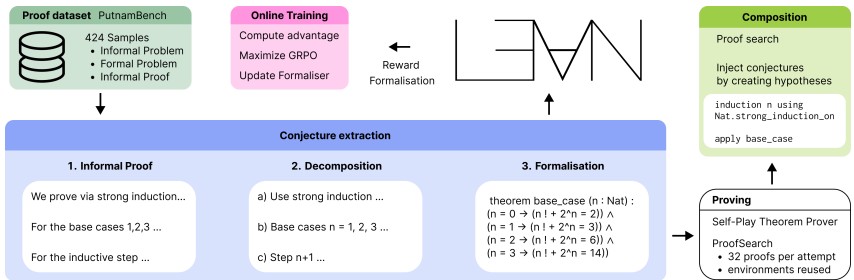

Figure 1: A general overview of conjecture extraction.

- We open-source a proof-search implementation combining ATP, autoformalization, and conjecture generation with online reinforcement learning for ATP and autoformalization.
- We provide initial experiments, showing the potential of conjecture extraction, improving absolute 11.2% over DSP in pass@1 proof rate on MiniF2F test Zheng et al. (2022).

## 2 RELATED WORK

Our work builds upon existing methods in proof autoformalization, conjecture autoformalization, and ATP, with a focus on the Lean 4 proof assistant Moura & Ullrich (2021).

**Proof autoformalization**  Proof autoformalization is translating an existing informal proof into a formal one Wang et al. (2024). A prominent method is DSP, where an LLM generates a high-level proof sketch from an informal proof, leaving smaller steps to be solved by pre-existing ATP methods. Originally designed for the Isabelle proof assistant, Lin et al. (2025) report problems porting DSP into Lean 4. Aniva et al. (2025) successfully port DSP into Lean 4. But as LLMs struggle to adhere to consistently adhere to Lean 4 syntax, the performance of DSP in Lean remains limited Aniva et al. (2025). Another limitation of DSP is its incompatibility with modern whole-proof generation models Wang et al. (2025), as it requires filling in an existing proof structure Jiang et al. (2023).

**ATP**  LLM-based ATP largely follows two paradigms: proof-step generation and whole-proof generation. Proof-step generation is an interactive process where a model predicts the next tactic based on the current proof state Polu & Sutskever (2020), often enhanced with tree-search methods Polu et al. (2023). In contrast, whole-proof generation models produce an entire proof in a single completion Xin et al. (2024). Due to greater computational efficiency, whole-proof generation has become the dominant approach in state-of-the-art ATP systems Wang et al. (2025).

**Conjecture generation**  The performance of modern ATP models relies on training via expert iteration, a process that requires large-scale datasets of formal conjectures. To overcome the scarcity of human-written formal problems, many works generate data by autoformalizing large corpora of informal mathematics, creating datasets like the Lean workbook. A more scalable approach is to generate entirely new conjectures. The effectiveness of this was demonstrated by Dong & Ma (2025), who trained a dedicated conjecture generation model and ran expert iteration on its outputs to achieve state-of-the-art ATP performance.

**LegoProver**  Wang et al. (2024) use conjecture generation for proof autoformalization for *Lego-Prover*, a work closely related to conjecture extraction. LegoProver uses Isabelle, where it aims to solve difficult problems by generating and maintaining a library of proven conjectures. It enhances the DSP framework by using a retrieval mechanism to select useful, already proven premises for a given proof attempt. These premises, along with their proofs, are then copied directly into the context of the LLM, which generates a proof sketch for the main theorem in a two-step process. The informal proof is first rewritten to explicitly state individual proof steps, before this rewritten informal proof is used for a proof-sketch. As LegoProver sketches similar to DSP, the approach is limited to proof-step generation models and does not generalize to whole-proof generation models.

Conjecture extraction also targets individual proof steps but formalizes each step into its own separate conjecture. After these smaller conjectures are proven, the full proof is recomposed. This method only requires the LLM for the natural language task of decomposing the proof, rather than prompting it to generate a complete, syntactically correct formal proof sketch.

**Benchmarks** Benchmarks for proof autoformalization are sparse. Only few ATP benchmarks also include informal proofs, enabling proof autoformalization evaluation. A popular ATP benchmark with informal proofs is MiniF2F Zheng et al. (2022). PutnamBench Tsoukalas et al. (2024), a famous ATP benchmark based on the William Lowell Putnam Mathematical Competition, only partially contains informal proofs. For our experiments, we web-scrape informal Putnam proofs of the Internet, leading to 424 samples with informal proofs. Other benchmarks, such as Lean workbook Ying et al. (2024), do not provide informal proofs. While ProofNet by Azerbayev et al. (2022) contains informal proofs, the Lean 4 port by Xin et al. (2024) does not. Therefore, our evaluations focus on MiniF2F and PutnamBench.

## 3 METHODOLOGY

This work's major contribution is *conjecture extraction*, which we cover first. Conjecture extraction requires conjecture autoformalization, and we discuss the two methods we tested next. We lastly end with brief implementation details, focusing on our open-source contribution.

**Conjecture extraction** Decomposing challenging proofs into easier subtasks has been proposed by Wang et al. (2024). Conjecture extraction achieves this by conditioning an LLM on a short prompt together with three problem-specific data points:

- the informal problem statement,
- an informal proof of the problem,
- and a formalized problem statement.

The LLM is tasked with extracting individual isolated proof steps as conjectures from the informal proof. We consider the initial conjecture used to extract conjectures the *parent conjecture*, whose *children* are generated using an LLM. These children adhere to a specified format to facilitate parsing. Specifically, a child conjecture consists of four parts:

1. a preceding *reasoning* block following the chain of thought paradigm
2. a *given* section which introduces variables and their ranges
3. an *assumes* block which states required hypotheses
4. a *shows* area with the conclusion that follows from the variables and assumptions

Note that the differentiation between variables and assumptions is rather blurry. For example, an integer $a$ introduced in the given section could be stated as `positive integer` or as `integer`. In the latter case, positivity could be ensured with an assumption $a > 0$. Both conjectures could yield the same formalization. Still, we decided to separate variables and assumptions, as we found that this pattern arises naturally when prompting various LLMs. We use this format to achieve lower perplexity prompts, which benefits overall generation quality Gonen et al. (2023). Once the conjectures are extracted, the LLM output is parsed and autoformalized.

**Autoformalization** To formalize conjectures, we test both a general-purpose model and a specialized model trained for autoformalization of competition problems Wang et al. (2025).

The existing general-purpose LLMs often generate Lean 3 code, as Lean 4 was released in 2021 and existing Lean code still frequently contains Lean 3 syntax. When zero-shot prompting the generation of Lean 4 code for natural language conjectures, we often faced syntactical errors in the output code containing a mix of Lean 3 and Lean 4 syntax. To mitigate this, we rely on in-context examples Dong et al. (2024).

For both models, following Poiroux et al. (2025), we generate multiple formalizations until the first type-checks. Specifically, we query the LLM to generate a formalization, pass it to the Lean REPL

LeanProver-Community (2025), and retry if the formalization is erroneous, as shown in Figure 2. We retry up to $F = 10$ times, balancing compute and success rates Poiroux et al. (2025). If no formalization is possible, we skip this natural language conjecture and exclude it from the rest of our pipeline.

We perform this formalization for each generated conjecture individually, parsing them using the format described above.

**Naming collision**   When formalizing many conjectures, we faced the issue of theorem name collisions. Multiple conjectures might receive the same name from our formalization component. If these conjectures are proven and loaded in the same Lean environment, Lean fails, as the name is already in use. To mitigate this, two possibilities arise.

1. Using a separate namespace for each formalized conjecture
2. Introducing a unique affix per theorem that is added to the name

Both variants work equally well and have their implementation challenges. Namespaces are Lean's native way to support the same names in different contexts. We therefore decided to use namespaces to resolve naming collisions. We introduce one namespace per theorem, assigning namespace names via UUID.

**Proving**   To prove each of the extracted conjectures, we rely on two pre-existing ATP methods. Initial experiments are performed with ReProver Tacgen, an encoder-decoder model for proof-step generation developed by Yang et al. (2023). ReProver supports premises natively, so it does not require the hypothesis injection described below, and thereby serves as a good starting point for our tests.

Later, we generate whole proofs with Self-Play Theorem Prover by Dong & Ma (2025) instead. For each conjecture, we sample $T = 32$ possible proofs from Self-Play Theorem Prover or $T = 32$ proof steps from ReProver, respectively. For both models, we enhance proof attempts with similar proven conjectures.

**Premise selection**   We follow Wang et al. (2024) in selecting the most relevant proven conjectures to facilitate further proving. When proving a conjecture, we embed its initial proof state using the ReProver retrieval encoder Yang et al. (2023) and retrieve the $k = 10$ nearest premises for it, measured by comparing the cosine similarity. Note that Self-Play Theorem Prover was not trained to deal with additional premises. Simply adding the premises before the theorem to be proven resulted in detrimental performance in our preliminary tests. To nonetheless inject premises, we consider three strategies:

1. Explicitly adding tactics such as `apply premise` and `rw [premise]` for every premise at each proof step. Although straightforward, this method significantly enlarges the search space, impacting efficiency and overall proof rates with the same compute.
2. Converting each premise into a **have** statement embedded within the proof for the overarching statement. This approach enables the LLM used in ATP to select the right premises by adding them to the prompt in a format usable even by whole-proof generation models. However, the resulting proofs are significantly longer, as each proof might contain up to $k$ additional sub-proofs of similar length. Lean only requires the existence of a term of a particular type, rather than the full proof given in the **have** statement.
3. Transforming proven premises into hypotheses by extracting their types and appending the types directly to the theorem statement as hypotheses, which we term *hypothesis injection*. This provides the essential information that such a proof term exists, without exposing unnecessary details.

We choose the third variant as it efficiently provides sufficient information for ATP tools to guide the search space by letting it choose when to apply hypotheses, without unnecessary lengthy proofs.

Internally, hypothesis injection uses the pretty-printed type of a declaration and injects this as Lean syntax in the theorem statement. Over all tests combined, we noticed one edge-case where using the pretty-printed string fails. Definite integrals $\int_a^b f(x)dx$, with the correct syntax in Lean as

$\int$ x **in** (a)..(b), f x are pretty-printed as $\int$ x **in** a ..b, f x. Note the missing parentheses and incorrect indentation.

**Proof checking**   We execute the generated theorem with its proof in the Lean REPL LeanProver-Community (2025). The proof-step generation interface is used for ReProver and the whole-proof generation interface for Self-Play Theorem Prover. If unsolved goals remain or an error is encountered, the proof candidate is considered a failure.

We store all proven conjectures in a Lean project and run `lake build` to guarantee the correctness in case of implementation errors in the Lean REPL. To utilize Lean's caching mechanism for faster build times, we place each conjecture in a separate file and import these into the project's main file.

If a conjecture could not be proven, it is re-enqueued for proof search. When next attempted, other new conjectures will have been proven and might be selected in the premise selection step, making the conjecture easier to prove.

**Hypothesis rejection**   Similarly to Xin et al. (2024), we use hypothesis rejection to filter conjectures with invalid assumptions before attempting a proof. We replace the conclusion of each conjecture with `False` and attempt a proof for `False` using the assumptions. If a proof is found, we discard the conjecture.

**Technical details**   We open-source an implementation merging conjecture generation, autoformalization, and ATP via message queueing. It supports both whole-proof generation and proof-step generation models, searching via HyperTreeProofSearch Lample et al. (2022), and online reinforcement learning for autoformalization and ATP through the interplay of various components, see Figure 3. With this, we are the first open-source implementation of online reinforcement learning for ATP and online reinforcement learning for autoformalization. Specific details for individual parts of the implementation can be found in the appendix.

While we evaluate conjecture extraction, the same implementation can also be used directly for online training of ATP models, without extracting any conjectures. Our evaluations show promising results, but should mostly be considered preliminary. We encourage researchers to explore more possibilities of the new pipeline, hoping our contribution will facilitate open-source AI for formalization.

# 4   EVALUATION

As conjecture extraction can be used with any ATP tool, we test it with hammers, proof-step, and whole-proof generation models.

**DSP comparison**   Starting the conjecture extraction evaluations, we test it using the same naive hammers as Aniva et al. (2025): `aesop`, `simp`, and `linarith`, as the only possible proof steps. For DSP, Aniva et al. (2025) similarly use GPT-4o Hurst et al. (2024) to generate one proof sketch per problem. Our setup therefore allows for direct comparison against their results with DSP in Lean.

Conjecture extraction **significantly outperforms DSP** as the current state of the art for proof autoformalization, achieving an **absolute increase of 11.2%** for proof rate on MiniF2F test. DSP errors primarily stem from the LLM not adhering to Lean 4 syntax, resulting in proof-sketch formalization failures. Remarkably, even a naive application of `aesop` to each conjecture without any LLM generation performs similar to DSP.

Subsequently, we compare conjecture extraction with a direct application of the ATP tool for proof-step and whole-proof generation models, discarding DSP due to the low proof sketch success ratio.

**Proof-step generation**   We test proof-step generation using ReProver Yang et al. (2023) on PutnamBench Tsoukalas et al. (2024). Conjecture extraction using ReProver Yang et al. (2023) **solves 2 PutnamBench problems**, **improving over the 0 problems by the default ReProver** Tsoukalas et al. (2024). The only better-performing proof-step generation models are InternLM2-StepProver Wu et al. (2024b) and InternLM2.5-StepProver Wu et al. (2024a).

Table 1: Proof rates of DSP, conjecture extraction, and the direct application of `aesop` on the MiniF2F benchmark Zheng et al. (2022). We sample one proof sketch and extract conjectures once, respectively.

| Method | Validation | Test |
|---|---|---|
| Aesop | 11.5% | 12.7% |
| Pantograph (DSP) | 12.7% | 14.7% |
| Conjecture extraction | **21.3**% | **25.9**% |

ReProver's parameter count is 300 million Xue et al. (2022), compared to 7 billion for both InternLM models Wu et al. (2024b). We highlight this difference in parameter counts in Figure 4. Importantly, **all proof-step models outperforming our approach are more than** $20\times$ **larger**. Beyond proof-step generation, conjecture extraction is the first proof autoformalization method to work on whole-proof generation, which we evaluate next.

**Whole-proof generation**  For whole-proof generation, we test conjecture extraction with Self-Play Theorem Prover Dong & Ma (2025) on MiniF2F Zheng et al. (2022). In our tests, Self-Play Theorem Prover without conjecture extraction proves 135 MiniF2F problems with pass@32. We extract conjectures five times and generate 32 proof candidates for each conjecture, creating a pass@$5 \times 32$ setting.

Figure 5 depicts the conjecture counts at the individual stages. 1192 conjectures are extracted and formalized within the five rounds, of which 200 are successfully proven. Twelve conjectures are rejected using hypothesis rejection. After five rounds of conjecture extraction, we cumulatively prove 144 MiniF2F problems. Our results indicate that **repeated conjecture extraction continuously boosts performance** of pre-existing whole-proof generation models, as shown in Figure 5. We attribute this to different natural language proof steps formed from the same informal proof and different formalizations of similar proof steps, which might be easier to prove.

## 5 CONCLUSION

In this work, we have presented *conjecture extraction*, an alternative proof autoformalization method that does not rely on proof sketches. Thereby, conjecture extraction can be used with whole-proof generation models. Our preliminary experiments demonstrate that conjecture extraction substantially outperforms DSP in Lean 4, achieving an 11.2% absolute improvement in the pass@1 proof rate on the MiniF2F test benchmark.

Iteratively extracting and proving conjectures leads to a continuous increase in the number of successfully proven theorems. We open-source an implementation of our pipeline, which combines conjecture generation, autoformalization, and ATP with online reinforcement learning, and can be used for large-scale training and proof autoformalization testing. By providing this framework, we hope to facilitate further research and development in the open-source AI community for formal mathematics. Ultimately, conjecture extraction represents a promising new direction for tackling the challenges of large-scale proof autoformalization, aimed at aiding mathematicians working on formalization projects.

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

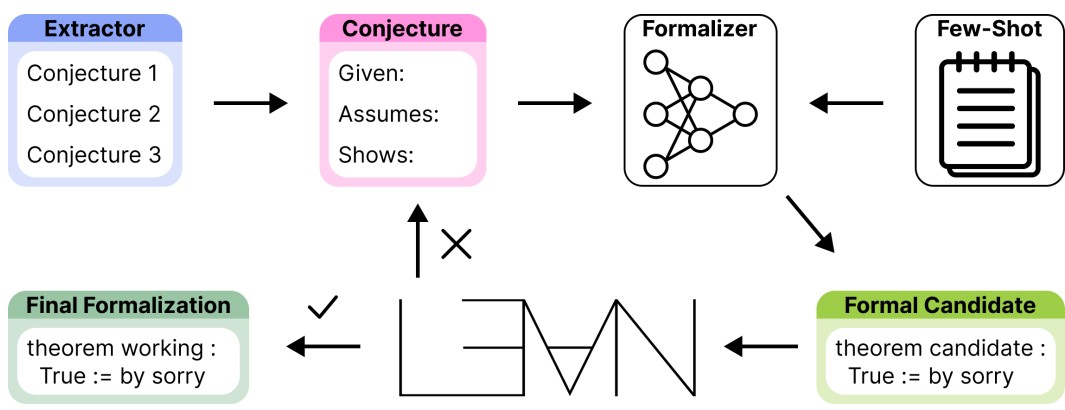

Figure 2: We formalize each conjecture using one-shot in-context learning Dong et al. (2024). Lean Moura & Ullrich (2021) type-checks the conjecture using the Lean REPL LeanProver-Community (2025). If this fails, we retry the formalization of the same conjecture up to 10 times.

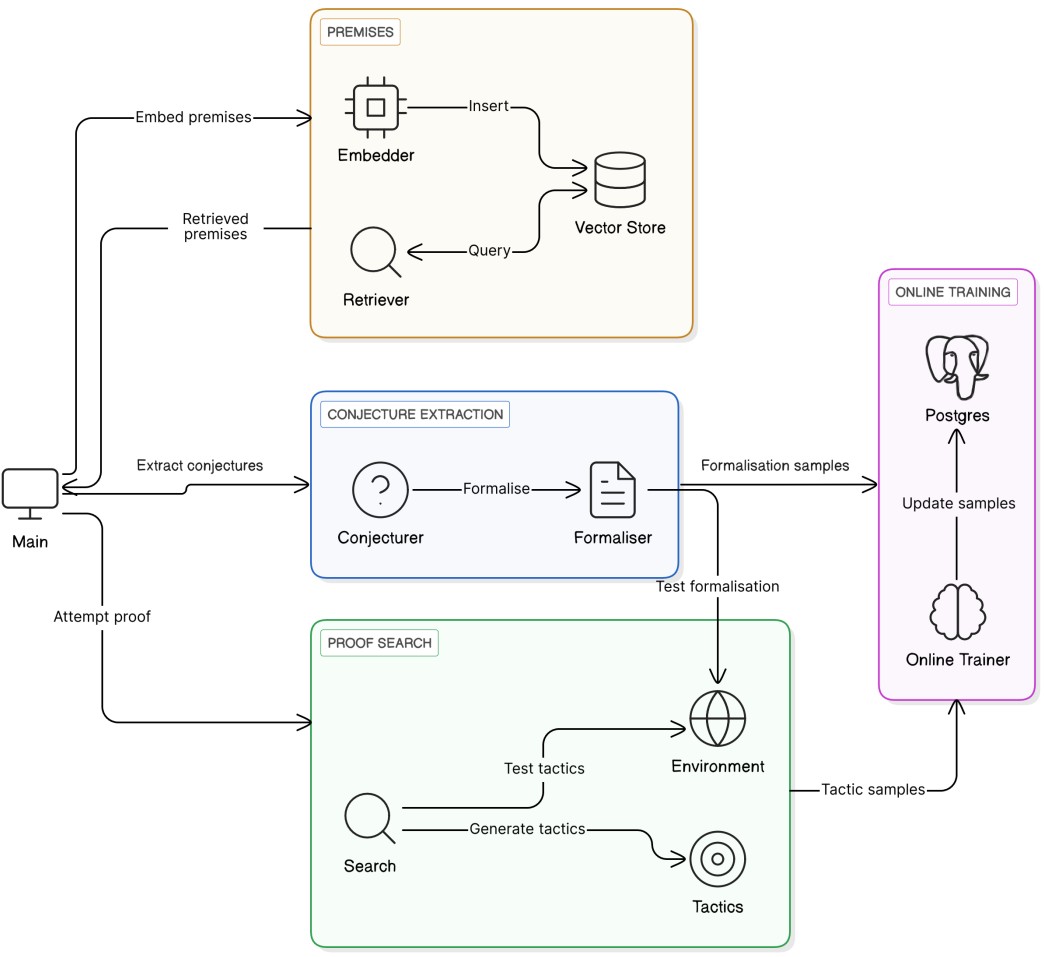

Figure 3: A simplified overview of the open-sourced implementation.

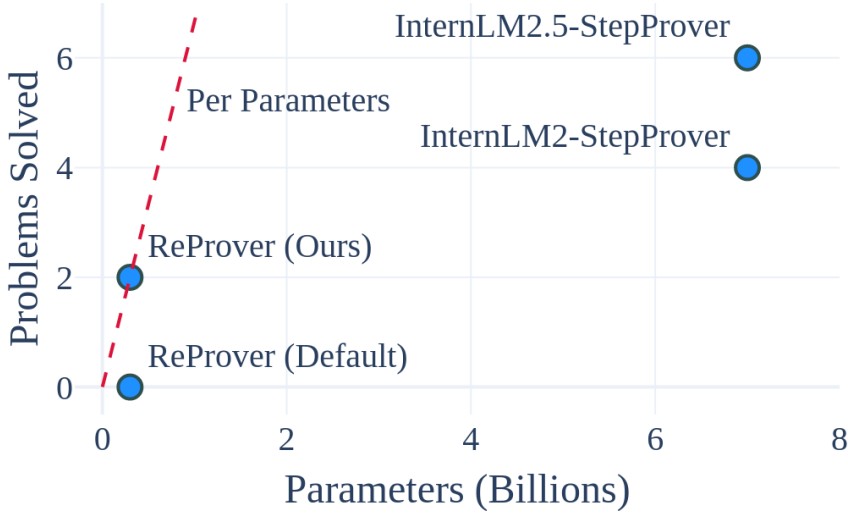

Figure 4: Proven Putnam problems and parameter counts for proof-step generation models

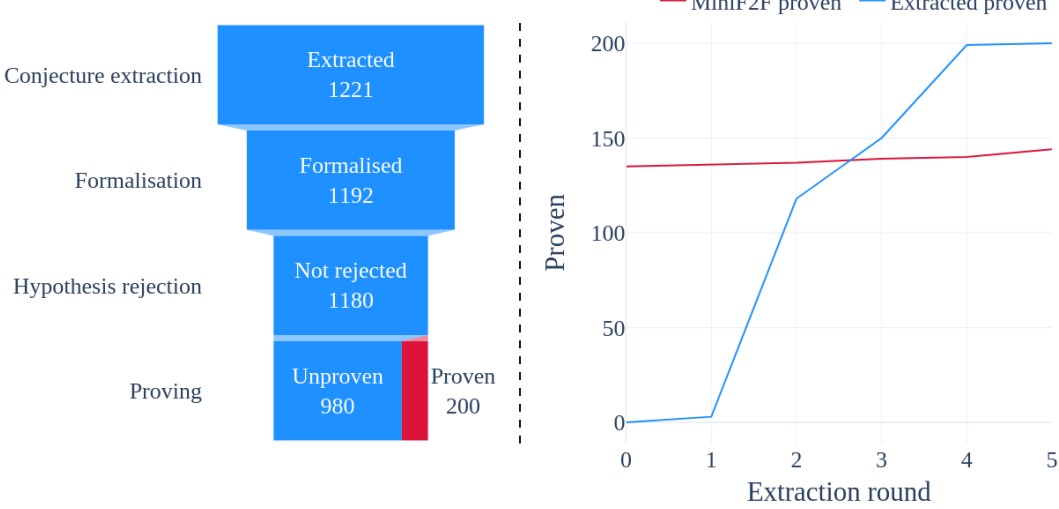

Figure 5: Left: conjecture counts per pipeline step. Right: more MiniF2F problems are proven with repeated conjecture extraction.

