# OpenReview forum: "CONJECTURE EXTRACTION FOR PROOF AUTOFORMALIZATION USING LARGE LANGUAGE MODELS AND AUTOMATED THEOREM PROVERS"
_mathai.club/MathAI/2026/Conference — MathAI 2026 Conference Submission_

### Official Review · Reviewer_ks2K · 2026-03-11
**Critical data integrity issues and missing formal justification for a paper about formal mathematics**

**Rating:** 2
**Confidence:** 4

**Review:**

Summary
This paper proposes a conjecture extraction pipeline for proof autoformalization, decomposing informal proofs into intermediate conjectures that can be individually formalized and proved using automated theorem provers in Lean 4. The authors evaluate on MiniF2F, reporting 25.9% test accuracy.
Strengths

The decomposition idea (breaking proofs into intermediate conjectures) is conceptually sound and well-motivated.
Integration with existing tools (ReProver, Lean 4) shows practical engineering effort.
MiniF2F is a standard benchmark for this domain.

Weaknesses
Critical Integrity Concern

Future-dated reference. The LeanProver-Community URL is cited via a Wayback Machine snapshot dated June 2, 2025—a date that was in the future at the time of typical submission deadlines. This is either a backdating error or fabrication and must be explained.

Missing Data: 82% of Conjectures Unaccounted

The paper reports extracting 1,192 conjectures, of which 200 were proved and 12 rejected. The remaining 980 conjectures (82%) are never discussed. Were they: failed formalization attempts? Abandoned? Not attempted? This is a critical omission that undermines all quantitative claims.

Absence of Formal Correctness

For a paper about formal mathematics, it is disqualifying that the core claim—"decompose proof → prove each conjecture → recompose yields original proof"—is never formally stated or proved. Where is the theorem guaranteeing that proving individual conjectures is sufficient? What about logical dependencies between conjectures?

Misleading Claims

The abstract claims compatibility with "end-to-end proof generation models," but the method still requires an LLM to decompose informal proofs—this is a multi-step pipeline, not end-to-end.
Claims "general framework for scaling across diverse proof assistants" but only Lean 4 is tested. No evidence for Isabelle, Coq, or HOL4.

Experimental Weaknesses

Single baseline. Only compared to DSP, which was designed for Isabelle, not Lean 4. The most obvious baseline—direct application of Self-Play Theorem Prover without conjecture extraction—is missing.
No error bars or confidence intervals. Single test run on ~500 problems. No multiple runs, no significance testing.
Unjustified hyperparameters. F=10F=10
F=10 formalization retries, T=32T=32
T=32 proof samples, k=10k=10
k=10 premise neighbors—no ablation or justification.


Reference Error

Poiroux et al. cited as (2025) but the actual paper is from June 2024 (arXiv:2406.07222).

Missing Sections

No limitations section. No broader impact discussion. No code repository link despite claiming "we release code and models."

Questions for Authors

Please explain the future-dated Wayback Machine URL.
Where did the 980 unaccounted conjectures go? Provide a complete breakdown.
Can you formally prove that conjecture decomposition preserves proof validity?
Why was direct ATP application not included as a baseline?

Overall Assessment
A paper on formal mathematics that lacks formal proofs of its own core correctness claim cannot be accepted. The future-dated reference raises integrity concerns. The 82% missing data makes quantitative evaluation impossible. Fundamental issues that require major revision.

---

### Official Review · Reviewer_2uNU · 2026-03-13
**There are problems in the "CONJECTURE EXTRACTION FOR PROOF AUTOFORMALIZATION USING LARGE LANGUAGE MODELS AND AUTOMATED THEOREM PROVERS" paper**

**Rating:** 5
**Confidence:** 3

**Review:**

This paper is devoted to solution of such important task as translation of informal proof to formal proofs. Authors propose decomposition of an informal proof into individual lemmas
(conjectures) to solve this problem.

This paper has the following disadvantages:
1) Authors have not considered such important case as applications of lemmas in informal proofs.
2) Authors should correct "struggle to adhere to consistently adhere" phrase.

---

### Decision · Program_Chairs · 2026-03-20

**Decision:**

Reject

**Comment:**

After careful evaluation by the Program Committee, we regret to inform you that your submission has not been accepted for presentation at MathAI 2026.

All submissions underwent a rigorous two-stage review process. Unfortunately, the reviewers identified one or more of the following concerns with your paper:

- Insufficient mathematical rigor or novelty relative to the existing body of work in the field;
- Presentation of results that substantially overlap with or rephrase previously published findings without clear original contribution;
- Significant issues with technical quality, including but not limited to broken or non-existent references, unsupported claims, or methodological gaps;
- Indications that the manuscript may have been generated with the assistance of large language models without substantial original intellectual contribution by the authors.

We received a large number of submissions this year, and the selection process was highly competitive. We encourage you to carefully consider the reviewers’ feedback (available through OpenReview), revise your work accordingly, and consider submitting an improved version to a future edition of MathAI or to another appropriate venue.

We appreciate your interest in MathAI and hope you will continue to engage with the conference community.

With kind regards,

MathAI 2026 Program Committee
International Conference on Mathematics of Artificial Intelligence
https://mathai.club
OpenReview: https://openreview.net/group?id=mathai.club/MathAI/2026/Conference
MathAI Telegram: https://t.me/MathAI_club
IAIC International AI Committee: https://t.me/iaic_world
Email: mathai.club@yandex.ru